# Systemic Therapy for HER2-Positive Metastatic Breast Cancer: Current and Future Trends

**DOI:** 10.3390/cancers15010051

**Published:** 2022-12-22

**Authors:** Kreina Sharela Vega Cano, David Humberto Marmolejo Castañeda, Santiago Escrivá-de-Romaní, Cristina Saura

**Affiliations:** 1Medical Oncology Department, Vall d’Hebron University Hospital, 08035 Barcelona, Spain; 2Breast Cancer Group, Vall d’Hebron Institute of Oncology (VHIO), 08035 Barcelona, Spain

**Keywords:** HER2 positive, metastatic breast cancer, targeted therapies, trastuzumab, antibody-drug conjugates, trastuzumab deruxtecan, tyrosine kinase inhibitors

## Abstract

**Simple Summary:**

Therapeutic advances in the treatment of HER2-positive metastatic breast cancer have dramatically improved the natural history of these patients. Although double anti-HER2 blockade associated with a taxane currently remains the best option in the first line, recently, T-DXd has positioned itself as the new standard in the second line. Eventually, most patients progress and develop resistance. In response to this need, new treatments and combinations are being developed. The aim of this paper was to review the new strategies that are in development for the treatment of these patients, the mechanisms of action and toxicities. Treatments are expected in the near future to change the treatment paradigm.

**Abstract:**

Approximately 20% of breast cancers (BC) overexpress human epidermal growth factor receptor 2 (HER2). This subtype of BC is a clinically and biologically heterogeneous disease that was associated with an increased risk for the development of systemic and brain metastases and poor overall survival before anti-HER2 therapies were developed. The standard of care was dual blockade with trastuzumab and pertuzumab as first-line followed by TDM-1 as second-line. However, with the advent of new HER2-targeted monoclonal antibodies, tyrosine kinase inhibitors and antibody- drug conjugates, the clinical outcomes of patients with HER2-positive BC have changed dramatically in recent years, leading to a paradigm shift in the treatment of the disease. Notably, the development of new-generation ADCs has led to unprecedented results compared with T-DM1, currently establishing trastuzumab deruxtecan as a new standard of care in second-line. Despite the widespread availability of HER2-targeted therapies, patients with HER2-positive BC continue to face the challenges of disease progression, treatment resistance, and brain metastases. Response rate and overall life expectancy decrease with each additional line of treatment, and tumor heterogeneity remains an issue. In this review, we update the new-targeted therapeutic options for HER2-positive BC and highlight the future perspectives of treatment in this setting.

## 1. Introduction

Breast cancer (BC) is the most prevalent type of malignancy in females; approximately 12.9 percent of women will be diagnosed with breast cancer at some point during their lifetime [1]. BC is a very heterogeneous disease that can be divided into several subtypes with diverse clinical characteristics and different prognoses. Approximately 20% of all breast cancers have gene amplification or overexpression of human epidermal growth factor receptor 2 (HER2), a transmembrane receptor tyrosine kinase. These result in tumors with a more aggressive phenotype and poor prognosis; about 50% of these will also have expression of estrogen and/or progesterone receptors (ER/PR) [2].

HER2 is a membrane receptor with tyrosine kinase (TK) activity, a member of a family of four receptors: epidermal growth factor receptor (EGFR o HER1), HER2, HER3, and HER4. When ligand–receptor binding occurs in the extracellular domain, it promotes a conformational change that induces binding to another molecule of the family, conforming a dimer, and then activation of the intracellular TK domain. The four receptors can bind to each other, conforming homodimers or heterodimers; dimerization is essential for receptor activation. A peculiarity of HER2 is that it does not have a ligand-binding site, but its conformation is like ligand-binding EGFR, permanently dimerizable and, therefore, active. Once the receptor is activated, important metabolic pathways such as the mitogen-activated protein kinase (RAS/RAF/MEK/ERK) pathway and phosphatidylinositol 3-kinase (PI3K/AKT/mTOR) pathway are activated. These pathways are essential for cell survival, growth and differentiation [3]. 

The HER2 receptor and the whole HER2 signaling pathway have been studied as targets for cancer therapy. Both the upstream and downstream of the HER2 signaling pathway are promising targets to block the signal and inhibit tumor growth [4]. HER2 antibodies and antibody-drug conjugates (ADCs) recognize the extracellular domain of the receptor. Both prevent the dimerization and consequently the activation of intracellular pathways. At the same time, they stimulate the antibody-dependent immune system. On the other hand, the tyrosine kinase inhibitors (TKIs) recognize the receptor’s intracellular domain and inhibit the oncogenic signaling at this point [5].

Currently, in the first-line setting, regardless of hormone receptor (HR) status (ER/PR), the standard treatment was established by the CLEOPATRA trial, which demonstrated that adding pertuzumab to docetaxel and trastuzumab significantly increased median progression-free survival (mPFS) (18.5 versus 12.4 months with and without pertuzumab, respectively, HR 0.62; 95% CI 0.51–0.75 months; *p* < 0.001) [6]. At a median follow-up of >8 years, there was a 16.3 month improvement in median overall survival (mOS) (57.1 months versus 40.8 months, HR 0.69; 95% CI 0.58–0.82 months) in favor of double anti-HER2 blockade. If tolerated, docetaxel or paclitaxel is recommended for at least six cycles, followed by maintenance trastuzumab-pertuzumab until progression [7]. For those HER2-positive, HR-positive tumors, endocrine treatment (ET) should be added to maintenance therapy [8].

Upon progression, the standard second-line treatment was trastuzumab emtansine (T-DM1); as determined by the results of the EMILIA trial, this clinical trial compared T-DM1 vs. lapatinib plus capecitabine in patients previously treated with trastuzumab and a taxane. T-DM1 demonstrated an improvement in mPFS of 3.6 months (HR 0.65, 95% CI 0.55–0.77; *p* < 0.001) and in mOS of 5.8 months (HR 0.68, 95% CI 0.55–0.85; *p* < 0.001) [9]. Recently, data from the DESTINY-Breast-03 trial have revealed a real “practice changing” clinical trial. These results demonstrated trastuzumab deruxtecan (T-DXd) superiority compared to T-DM1 in second line, positioning T-DXd as a new standard of care after the progression to trastuzumab and a taxane [10]. So far, the choice of the third line and successive treatments does not have a standard; guidelines and experts recommend maintaining anti-HER2 therapy such as trastuzumab or lapatinib combinations with chemotherapy or endocrine therapy, tucatinib/trastuzumab/capecitabine, neratinib/capecitabine and T-DXd or T-DM1 if not previously used [11,12].

Although HER2-targeted therapy has changed the natural history of HER2+ MBC, the disease eventually progresses in most cases, highlighting the need for new targeted therapies for advanced disease. In this review, we look for the evidence derived from the most promising strategies that may become future therapeutic options and explain the mechanisms of action and the results reported so far with these new approaches to treat HER2+ MBC. Finally, we list some of the ongoing clinical trials that may become a part of future therapeutic options.

## 2. Novel Anti-HER2 Approaches

### 2.1. Antibodies

One of the new strategies in anti-HER2 treatment is modifying the constant fragment (Fc) chain of the monoclonal antibody that targets the HER2 receptor to enhance the antitumor activity based on cytotoxic natural killer (NK) cell recruitment [13], as developed in margetuximab. Another new anti-HER2/3 approach includes bi-specific antibodies, such as Zanidatamab.

#### 2.1.1. Margetuximab

Margetuximab is an antibody that shares the same variable fraction as trastuzumab and, therefore, binds in the same epitope of the HER2 receptor, but incorporates an engineered Fc region to increase immune activation in order to improve the antibody-dependent cellular cytotoxicity (ADCC) favoring better recognition of the tumor by NK cells.

The NK cells and macrophages express on their surface the stimulatory receptor FcγRIIIA (CD16A) that triggers ADCC by innate immune cells. This receptor is encoded by two alleles that differ in the codon for amino acid 158 conforming two polymorphisms of CD16A: a V-variant with higher affinity (valine [V]) and an F-variant with lower affinity (phenylalanine [F]). The clinical benefit of therapeutic antibodies appears to be greater for patients with the VV genotype compared to those with the lower-affinity FV and FF genotypes (CD16A-158F carriers). Approximately 85% of individuals are carriers of the CD16A-158F allele. Modifications have been made to the IgG1 Fc of margetuximab, five amino acid substitutions that result in increased binding to both CD16A isoforms and reduced binding to CD32B, an FcγR inhibitor [14]. In the phase 1 study, 66 patients received margetuximab; the treatment was well-tolerated, with mostly grade 1 and 2 toxicities such as pyrexia, nausea, anemia, diarrhea and fatigue. Despite the heavily pre-treated population (three median prior therapies), margetuximab demonstrated evidence of clinical activity [15].

In December 2020, margetuximab was approved by the FDA on the basis of the results of the SOPHIA study. The SOPHIA phase 3 trial compared the clinical efficacy of margetuximab vs. trastuzumab, each with chemotherapy of choice of the investigator among predefined options (capecitabine, vinorelbine, eribulin or gemcitabine), in patients with pretreated HER2-positive MBC. A total of 536 patients were enrolled with disease progression on two or more prior anti-HER2 therapies and one to three lines of therapy for metastatic disease (91.2% had received prior T-DM1). Median PFS in the margetuximab arm was 5.8 months compared with 4.9 months in the control arm (HR 0.76; 95% CI: 0.59, 0.98; *p* = 0.033). The objective response rate (ORR) was 22% with a median duration of response (DoR) of 6.1 months in the margetuximab arm compared to an ORR of 16% and median DoR of 6.0 months in the control arm [16].

Final OS results did not demonstrate a statistically significant advantage for margetuximab over trastuzumab (HR 0.95; 95% CI: 0.77–1.17; *p* = 0.62); the median OS was 21.6 vs. 21.9 months for patients treated with margetuximab vs. trastuzumab plus chemotherapy, respectively. Interestingly, in the subgroup analysis of the SOPHIA study, the OS was improved for the margetuximab plus chemotherapy arm in patients carrying a CD16A-158F lower affinity allele, whereas the high-affinity subgroup of patients homozygous for CD16-158V had improved OS with trastuzumab plus chemotherapy [16]. A limitation of the study is that patients were not stratified according to CD16A status, with the lack of a companion diagnostic test developed to select patients with the CD16A-158F allele that might yield the most benefit from margetuximab.

#### 2.1.2. Zanidatamab (ZW25)

Zanidatamab is a humanized, bispecific, immunoglobulin G isotype 1 (IgG1)-like antibody that binds simultaneously to two distinct sites on HER2, the juxta membrane domain (ECD4) and the dimerization domain (ECD2) of HER2. This design results in multiple mechanisms of action, including dual blockade of HER2 signaling, increased antibody binding, receptor clustering and removal of HER2 from the cell surface, and potent effector function.

In the phase 1 dose-escalation study of zanidatamab, ZW25 was administered weekly (QW; 5, 10, or 15 mg/kg) or biweekly (Q2W; 20 mg/kg) in 4 week cycles. Sixteen patients were included, 50% with breast cancer, all of whom had received trastuzumab and T-DM1 in previous lines. The most frequent AEs were diarrhea, infusion reaction, and fatigue. In seven breast cancer patients, two showed partial responses (PRs), two had stable disease (SDs) > 6 months, and three had progressive diseases (PDs) [17].

The promising single-agent activity prompted exploring the combination of ZW25 with chemotherapy. Twenty patients who had previously received trastuzumab, pertuzumab, and T-DM1 were included in the phase I trial. Zanidatamab (20 mg/kg Q2W or 30 mg/kg Q3W) was administered in combination with paclitaxel, capecitabine, or vinorelbine. The median number of prior systemic therapies was two regimens in the metastatic setting. The ORR was 37.5%, and the disease control rate was 81.3%. The most frequent AEs were diarrhea, nausea, peripheral neuropathy, and fatigue. The results of this study support further investigation of zanidatamab combinations with single-agent chemotherapy [18].

Other studies with different combinations are ongoing, such as ZW25 with taxanes in first-line metastatic patients; the preliminary results of this phase 1b/2 study (NCT04276493) were presented at ASCO 2022. The confirmed ORR was 86.4%, and the 6 month PFS rate was 90.9%. The most common adverse events were diarrhea (56.0%) and neutropenia (52.0%) [19]. Another ongoing trial is exploring the combination of ZW25 + palbociclib + fulvestrant in HER2-positive/HR-positive populations (NCT04224272), with results to be reported at the San Antonio Breast Cancer Symposium in 2022.

### 2.2. Anti-HER2 Antibody–Drug Conjugates (ADC)

Antibody-drug conjugates are targeted agents that link a cytotoxic drug (also known as payload) to a monoclonal antibody. The antibody recognizes an antigen on the tumor and releases the cytotoxic drug into the cell, thereby reducing toxicity and improving efficacy [20]. The cytotoxic drug and the antibody are joined by a connector (linker). The linker is a critical component in the structure of ADCs; there are two types, “cleavable” and “non-cleavable”. Linkers have to remain stable in circulation and be easily cleavable in the intracellular space to deliver the payload. “Cleavable” linkers release the drug by hydrolysis or proteolysis. “Non-cleavable” linkers depend on degradation of the antibody to release their cytotoxic payload. The average number of drugs conjugated per antibody is known as the drug-to-antibody ratio (DAR), and is an important feature of ADCs since it is relevant for potential efficacy and toxicity [21].

The ADC–antigen complex is usually taken up by endocytosis or pinocytosis after binding to its target. An early endosome is created as a result of internalization by budding inward from the cell membrane. Before binding to lysosomes, early endosomes become late endosomes [21]. Cleavage processes of ADCs with cleavable linkers, such as hydrolysis, proteolytic cleavage or reductive cleavage, take place in early or late endosomes, but ADCs with non-cleavable linkers need sophisticated proteolytic cleavage by cathepsin B and plasmin in lysosomes. The cytotoxic payload is discharged from lysosomes to the cytoplasm after cleavage. Cytotoxic payloads target DNA and cause single- or double-strand DNA breaks, or interfere with microtubules, leading to cell apoptosis [22].

#### 2.2.1. Trastuzumab Emtansine

Trastuzumab emtansine (T-DM1) is an ADC, in which trastuzumab is conjugated to microtubule inhibitor agent DM1 (derivative of maytansine); both are conjugated by means of a stable linker. An average of 3.5 DM1 molecules are conjugated to each molecule of trastuzumab.

This ADC has the mechanisms of action of both trastuzumab and DM1, and like trastuzumab, binds to domain IV of the HER2 extracellular domain, as well as to Fcγ receptors and complement C1q. In addition, it inhibits signaling through the phosphatidylinositol 3-kinase (PI3K) pathway and mediates antibody-dependent cell-mediated cytotoxicity (ADCC). DM1, the cytotoxic component, inhibits tubulin polymerization; it causes cells to arrest in the G2/M phase of the cell cycle, ultimately leading to apoptotic cell death.

The phase III clinical trial, EMILIA, led to the approval of T-DM1 by the FDA in 2013 as a second-line treatment for patients previously treated with trastuzumab and a taxane. This trial showed an improved mPFS of 9.6 months vs. 6 months (HR 0.65, 95% CI 0.55–0.77; *p* < 0.001), as well as a median overall survival (mOS) improvement of 30.9 vs. 25.1 months (HR 0.68, 95% CI 0.55–0.85; *p* < 0.001) for TDM1 vs. lapatinib plus capecitabine, respectively. The objective response rate for T-DM1 was 43.6% vs. 30.8% for the control arm (*p* < 0.001). In terms of adverse effects, most were grade 1 and 2. Thrombocytopenia (12.9%) and elevated transaminases (AST in 4.3% and ALT in 2.9%) were the most frequent G3 events associated with T-DM1 [9]. The clinical trial modified its protocol and allowed crossover to TDM-1. Twenty-seven percent of patients crossed over to trastuzumab emtansine. In the final analysis, the primary endpoint of the EMILIA study could be confirmed, and an improvement in mOS was demonstrated, of 29.9 months vs. 25.9 months (HR 0.75, 95% CI 0.64–0.88) for the T-DM1 arm with respect to the control. The safety profile was unchanged, so safety and efficacy were confirmed. [23]

T-DM1 was also compared with the treatment of physician’s choice in the phase III TH3RESA study in HER2-positive advanced breast cancer patients heavily pretreated with a median of four previous regimens for advanced disease, including trastuzumab and lapatinib, a taxane in any setting and with progression on two or more HER2-directed regimens. This trial showed an improved mPFS of 6.2 vs. 3.3 months (HR 0.53, 95% CI 0.42–0.66; *p* < 0.0001) and a mOS of 22.7 vs. 15.8 months (HR 0.68, 95% CI 0.54–0.85; *p* = 0.0007) [24,25]. Results from the EMILIA and TH3RESA studies demonstrated that T-DM1 improves progression-free survival and overall survival in patients with advanced breast cancer refractory to other HER2-targeted treatments, with a favorable safety profile. For these reasons, until 2021, it remained as the standard of care in the second line for patients with HER2–positive MBC.

After its approval as a second-line standard treatment in the metastatic setting after progression to trastuzumab and a taxane, the phase III clinical trial KATHERINE compared TDM1 as adjuvant vs. trastuzumab. Postoperative systemic treatment, until that point, was trastuzumab for one year irrespective of pathological findings [26].

Only 40–60% of patients achieve complete pathologic response (pCR) with neoadjuvant treatment, and those with residual invasive disease have a higher risk of recurrence. This study demonstrated that TDM1 decreases the risk of recurrence of invasive disease and death from breast cancer by 50% in patients who did not achieve pCR after neoadjuvant treatment with a taxane and trastuzumab. At 3 years, 88.3% of patients in the T-DM1 group were free of invasive disease compared to 77% in the trastuzumab arm. Distant recurrence was reported in 10.5% of T-DM1 patients versus 15.9% of trastuzumab patients (HR = 0.60; 95% CI, 0.45 to 0.79) [26].

Regarding CNS recurrence, there was a higher incidence in the TDM1 arm compared to trastuzumab; however, the data suggest that this is the result of competing risk, as there was a reduction in recurrence outside the CNS as a first event. The cumulative incidence at any time point was similar in both arms; moreover, patients in the TDM1 arm had a longer median time to CNS recurrence, 17.5 months versus 11.9 months compared with trastuzumab [27].

The next challenge was to prove the efficacy and safety of T-DM1 as a first line. For this purpose, the MARIANNE study was designed, a phase III clinical trial with three arms: trastuzumab plus taxane as the control arm and two experimental arms, T-DM1 and T-DM1 plus pertuzumab. When the study was designed, in 2009 the combination of taxane with trastuzumab was the most commonly used first-line regimen, so one of the inconveniences of this study was that TDM1 was not directly compared with what is now the standard treatment, dual anti-HER2 blockade and taxane. None of the experimental arms demonstrated superiority in PFS. T-DM1 and T-DM1 plus pertuzumab demonstrated non-inferiority in PFS compared to the control arm, 14.1 months with T-DM1 and 15.2 months with T-DM1/pertuzumab versus 13.7 months with trastuzumab/taxane. Additionally, the response rate was lower than that of the control arm—67.9% in patients who had received trastuzumab plus taxane versus 59.7% and 64.2% in the T-DM1 and T-DM1/pertuzumab arms, respectively. The frequency and grade of adverse effects corresponded with those already seen in previous studies [28]. 

The development of a new generation ADCs with activity beyond progression or in patients refractory to T-DM1 and even in HER2-low tumors has become of great interest.

#### 2.2.2. Trastuzumab Deruxtecan

Trastuzumab deruxtecan (T-DXd o DS-8201) is another HER2-targeted antibody-drug conjugate. The antibody is a humanized anti-HER2 IgG1 that has the same amino acid sequence as trastuzumab, attached to deruxtecan, a topoisomerase I inhibitor (DXd) bound by a tetrapeptide-based cleavable linker [29]. Deruxtecan is an exatecan derivative, and it is 10 times more potent than the active metabolite of irinotecan. Trastuzumab deruxtecan has a DAR of 8, and it is a very stable drug in plasma. The antibody attaches to the HER2 receptors on the cell surface. This complex is internalized by the cell, and lysosomal enzymes cleave the linker. DXd is released, causing DNA damage and apoptosis. Moreover, the antibody portion has additional functions; it binds to FcγRIIIa and complement C1q and also inhibits signaling of intracellular metabolic pathways [29].

From the earliest studies, T-DXd demonstrated strong activity in HER2-positive MBC patients. It was initially tested in heavily pretreated patients. The DESTINY-Breast01 study, a single-arm phase 2 study, demonstrated results that led to worldwide regulatory approvals [30]. In that study, patients who had received prior treatment with TDM1 were included; the enrolled patients had received a median of six prior treatments. With a median duration of follow-up of 11.1 months, the ORR to T-DXd was 60.9%, and the mPFS was 16.4 months. The most frequent grade 3 or higher adverse events were neutropenia (20.7%), anemia (8.7%) and nausea (7.6%). Overall, 25 patients (13.6%) developed T-DXd-related interstitial lung disease (ILD). These events were mainly grade 1 or 2; however, there were four deaths (2.2% patients) attributed to ILD [28]. With additional follow-up, T-DXd still demonstrated durable efficacy, with a median duration of response of 20.8 months and 18 month overall survival of 74%, with a safety profile consistent with previous results [31].

After Destiny-01′s results, T-DXd was evaluated head-to-head vs. T-DM1 in the phase III DESTINY-Breast03 trial in HER2-positive MBC patients previously treated with a taxane and trastuzumab [10]. Striking efficacy was seen with T-DXd; the 12 month PFS rate was 75.8% with T-DXd versus 34.1% with T-DM1 (HR 0.28; 95% CI 0.22–037). The mPFS was not reached in the T-DXd group (95% CI, 18.5 to NE) and was 6.8 months in the TDM1 group (95% CI, 5.6 to 8.2). A strong trend in favor of OS benefit was also observed; the 12 month OS rate for T-DXd was 94.1% vs. 85.9% for T-DM1 (HR 0.56; *p* 0.007172), non-statistically significant since OS outcomes were not yet mature. The ORR was better in the T-DXd arm at 79.7% vs. 34.2% in the control arm (*p* < 0.0001). Sixteen percent of the patients in the experimental arm achieved a complete response, and partial responses were observed in 63.6% [10].

The safety profile was comparable between the two arms. T-DXd was associated with higher rates of nausea and vomiting without primary prophylaxis (72.8% any grade nausea; 6.6% grade > 3; 44% any grade vomiting; 1.1% grade > 3). The most common AE associated with treatment discontinuation for T-DXd was interstitial lung disease (ILD)/pneumonitis (8.2%), and for T-DM1, was thrombocytopenia (2.7%). The most common AEs associated with dose reduction for T-DXd were nausea (6.2%) and neutropenia (3.5%), and for T-DM1, were thrombocytopenia (4.2%) and ALT/AST increased (2.7% each). DESTINY 03 has been a practice-changing study and firmly establishes T-DXd in the second-line setting in patients with HER2-positive metastatic breast cancer [10,12]. 

Patients with treated and stable brain metastases (BM) were allowed to enroll in DESTINY-Breas03. In a subset analysis, an overall PFS benefit was observed in these patients in favor of T-DXd over T-DM1(HR = 0.25, 95% CI = 0.13–0.45). Central nervous system (CNS) objective response rates were also higher in patients treated with T-DXd (63.9% vs. 32.8%). However, the response data included only 36 patients treated with T-DXd, all of whom had previously treated, stable, and asymptomatic BM at baseline, and there was no reporting of time from most recent CNS-directed radiotherapy, which could have contributed to the responses observed [32]. 

In this setting, we have to mention two studies, DEBBRAH and TUXEDO-1. Both are yielding important scientific evidence on patients with progression at CNS level with active disease. The DEBBRAH trial (NCT04420598), is a phase 2 study evaluating T-DXd in patients with HER2-positive or HER2-low expressing (HER2-LE) advanced breast cancer and central nervous system (CNS) involvement, either brain metastasis or leptomeningeal carcinomatosis. This ongoing clinical trial enrolled patients into five cohorts, having reported results from three of them. Cohorts 1 and 3 included patients with HER2-positive breast cancer after CNS local therapy, whereas cohort 2 was for a population of asymptomatic untreated BM patients. For cohort 1 (n = 8), patients with non-progressing asymptomatic stable BM, the primary endpoint was PFS at 16 weeks. Results from this cohort demonstrated that seven of eight patients were non-progressing at 16 weeks, and the PFS rate was 87.5% (95% CI, 47.3–99.7; *p* < 0.001). Cohort 3 (n = 9) included patients progressing after local therapy; the primary objective was intracranial ORR, which was 44.4% (95% CI, 13.7–78.8; *p* < 0.001). Interestingly, cohort 2 (n = 4) showed an intracranial ORR of 50.0% (95%CI, 6.7–93.2) in patients with no previous CNS local therapy. Regarding safety, the most common grade 3 and 4 side effects were neutropenia and fatigue, respectively. G1 pneumonitis was reported in two patients [33]. Results for these cohorts demonstrated T-DXd preliminary efficacy with manageable toxicity in patients with stable and active BM. 

The phase 2 TUXEDO-1 trial (NCT04752059) included 15 patients with HER2-positive MBC and untreated central nervous system involvement (n = 6) or progression after prior local therapy (n = 9). The primary endpoint was the intracranial response rate. The study obtained positive results, with complete intracranial response in 13.3% of patients, partial intracranial response in 60%, and stable disease in 20%. The overall intracranial response rate was 73.3% (95% CI, 48.1–89.1%) with an mPFS of 14 months (95% CI, 11.0 months to unrecorded). Notably, a post hoc subgroup analysis shows a 100% RR in patients with de novo brain metastases, compared with 66.7% in patients with brain metastases progressing after previous local therapy. Safety was consistent with previous reports; regarding AEs of special interest, grade 2 interstitial lung disease and a symptomatic drop in LVEF were observed in one patient each. This study also showed promising results for T-DXd to treat patients with active BM, maintaining quality-of-life and neurocognitive functioning [34]. The ongoing DESTINY-Breast12 trial (NCT04739761) is enrolling up to 250 patients with either active or stable brain metastases and should shed further light on the role of T-DXd in patients with CNS disease.

With the encouraging results obtained to date in the second line, other ongoing trials are examining T-DXd in an even earlier-line setting. DESTINY-Breast07, for example, will investigate the safety, tolerability, and antitumor activity of trastuzumab deruxtecan in combination with other anti-cancer agents in the first-line metastatic setting [35]. In addition, DESTINY-Breast09 (NCT04784715), a phase 3 clinical trial, has been designed to prove the efficacy of T-DXd as a first line of treatment. In the study, patients are randomized to three arms: T-DXd and T DXd plus pertuzumab compared to the standard of care, the dual anti-HER2 blockade plus taxane [36].

The clinical benefit of T-DXd is also being evaluated in early breast cancer. Patients with HER2-positive early breast cancer with residual invasive disease after completing neoadjuvant treatment have a greater risk of disease recurrence or death and, currently, the standard of care is T-DM1. DESTINY-Breast05 (NCT04622319) is a phase 3 study to evaluate DXd vs. T-DM1 in patients with high-risk HER2-positive primary breast cancer who have residual invasive disease in breast or axillary lymph nodes following neoadjuvant therapy. 

#### 2.2.3. Trastuzumab Duocarmazine

Trastuzumab duocarmazine (SYD985) is a second-generation ADC comprising trastuzumab linked via a cleavable linker to the payload, a duocarmycin prodrug (vc-seco-DUBA). The DAR is 2.8. Proteolytic cleavage of the linker in the tumor microenvironment leads to activation of the prodrug payload. Its active toxin (DUBA) alkylates DNA and kills dividing and non-dividing cells. 

After demonstrating promising results in the phase I trial that included cohorts of HER2-positive and HER2-low patients [37], the phase III SYD985.002/TULIP compared trastuzumab duocarmazine with the physician’s treatment choice (PTC) in HER2-positive MBC patients who had received at least two prior lines of treatment or previous treatment with T-DM1 [38].

In this trial, the median number of prior treatments was 4 (range, 1–16) in the experimental group and 5 in the physician’s choice group. The study’s primary endpoint, blinded centrally reviewed median PFS, was 7.0 months for SYD985 and 4.9 months for PTC (HR 0.64 [0.49–0.84], *p* = 0.002). Median OS was 20.4 months with trastuzumab duocarmazine and 16.3 months with PTC, but the difference was not statistically significant at the reporting time point [38]. The overall response rate was similar between the trastuzumab duocarmazine and chemotherapy groups, at 27.8% versus 29.5%. 

Overall, adverse events of grade 3 or higher were reported in 52.8% and 48.2% of the trastuzumab duocarmazine and chemotherapy arms, respectively. The most common event of this grade was ocular toxicity in 21.2% of patients in the trastuzumab duocarmazine arm, which led to treatment discontinuation in 20.8% of patients and dose modifications in 22.9%. An additional 2.4% of patients developed interstitial lung disease or pneumonitis of this grade, which was associated with discontinuations and modifications in a corresponding 5.2% and 2.1% [38]. In light of these data, trastuzumab duocarmazine can provide a new treatment option for previously treated patients with HER2-positive metastatic breast cancer. Strategies are being developed to mitigate the toxic effects of this ADC, and the ongoing BYON5667.002 clinical trial (NCT04983238) is evaluating the safety and efficacy of sodium thiosulfate eye drops to reduce ocular toxicity in patients treated with SYD985.

#### 2.2.4. ARX788

ARX788 is an ADC that consists of a HER2 receptor-targeting monoclonal antibody linked to the cytotoxic payload AS269, a highly potent tubulin inhibitor. The antibody is different from trastuzumab; it incorporates a non-natural amino acid into the predetermined site on the heavy chain of the antibody, and AS269 is specifically conjugated to the non-natural amino acid. It has a DAR of 1.9 [39,40]. 

The results obtained in two phase 1 studies showed that ARX788 is active in HER2-positive, HER2-low and T-DM1-resistant tumors. The ACE-Pan tumor-01 trial, performed in HER2-positive solid tumors in the US and Australia, achieved an ORR of 67% (two out of three breast cancer patients) [41], and the ACE-Breast-01 trial, performed in HER2-positive breast cancers in China, achieved an ORR of 74% (14 out of 19 patients) [42]. With these results, a phase 2 clinical trial, ACE-Breast-03, a single-arm study in patients with HER2-positive breast cancer resistant or refractory to T-DM1, T-DXd and prior treatment with tucatinib, was ongoing; however, the pharmaceutical company will no longer directly pursue ARX788 and will pause clinical trials that include ARX788 outside of China and seek a development partner ex-China.

#### 2.2.5. ZW49

Zanidatamab zovodotin (ZW49) is a novel HER2-targeting bispecific ADC, composed of an antibody directed against two HER2 epitopes (non-overlapping) linked to an auristatin toxin with a protease-cleavable linker. The results of an ongoing phase 1 study with this novel ADC were recently presented at ESMO 2022. A total of 76 patients with previously treated HER2-positive cancers (51 in a dose escalation cohort and 25 in a dose expansion cohort) were treated with intravenous ZW49. Primary endpoints were safety and tolerability, and secondary endpoints included response assessments [43].

Different dose escalation schedules have been tested, reaching a recommended dose of 2.5 mg/kg for the Q3W schedule, with one dose-limiting toxicity of grade 2 keratitis reported. A total of 68 patients (89%) had treatment-related adverse events, mainly grade 1–2 keratitis (42%), alopecia (25%), and diarrhea (21%). Adverse events of grade 3 or higher were reported in seven patients (9%) (two grade 4 events in two patients, infusion-related reaction and decreased neutrophil count, respectively). From 29 response-evaluable patients treated with Q3W scheduled across different tumors, the confirmed objective response rate was 28%, and disease control rate was 72%. Enrollment is ongoing [43].

Currently ongoing trials with other HER2-targeted ADCs are listed in Table 1.

### 2.3. Tyrosine Kinase Inhibitors (TKIs)

Several HER1/HER2 TKIs, pan-HER TKIs, and dual HER2/VEGF TKIs are in different stages of clinical development.

#### 2.3.1. Lapatinib

The first TKI approved for the treatment of HER2-positive MBC was lapatinib, which inhibits HER2 and EGFR1. In the phase III EGF100151 trial, the combination of lapatinib plus capecitabine demonstrated a significant improvement in PFS (from 4.3 to 6.2 months, HR 0.55, 95% CI 0.40–0.74, *p* < 0.001) compared to capecitabine alone; also, a trend towards improvement in OS was observed. The trial was terminated early after preliminary analysis [44]. These results, previously the EMILIA trial, supported the use of this combination as a second-line regimen in patients with HER2-positive MBC after progression to trastuzumab treatment.

Another option, for patients with HER2-positive MBC whose disease has progressed on trastuzumab, is the combination of lapatinib and trastuzumab. The phase III EGF104900 trial demonstrated that the combination significantly improved PFS (HR 0.74, 95% CI 0.58–0.94, *p* = 0.011) and OS (HR, 0.74, 95% CI 0.57–0.97, *p* = 0.026) versus lapatinib monotherapy, offering a chemotherapy-free option for heavily pretreated patients. Associated grade 3 adverse events were mainly diarrhea, nausea, vomiting and hand-foot syndrome [45].

#### 2.3.2. Neratinib

An important member of this TKI family is neratinib, an irreversible pan-HER TKI (HER1, HER2, and HER4). 

In a phase II study that enrolled two cohorts, previous trastuzumab-based therapy and trastuzumab-naïve patients were treated with neratinib as a single agent. With mPFS of 22.3 and 39.6 weeks and ORR of 24% and 56% in each cohort, respectively, the study met its primary endpoint, which was a 16-week PFS of more than 30% (59% in pretreated group and 78% in naïve group) [46].

TBCRC-022 was a phase II trial that evaluated neratinib in combination with capecitabine, in two cohorts of patients (lapatinib-naïve or pre-treated-cohort 3A and 3B, respectively), with HER2-positive BC with CNS metastases. A total of 49 patients were enrolled (n = 37 in cohort 3A and n = 12 in cohort 3B). With a mPFS of 5.5 vs. 3.1 months and mOS of 13.3 vs. 15.1 months, respectively; the combination was highly active with a composite CNS ORR of 49% in cohort 3A and 33% in cohort 3B. Diarrhea was the most frequent common grade 3 toxicity in 29% in both cohorts [47].

The NALA clinical trial, a Phase III study, was designed to compare two TKIs, neratinib vs. lapatinib, both in combination with capecitabine in patients with metastatic HER2-positive breast cancer who had received at least two prior anti-HER2-based regimens. The study included patients with asymptomatic or stable CNS metastases (patients who had received local treatment or not) [48].

A total of 621 patients were randomized to receive neratinib in combination with capecitabine or lapatinib in combination with capecitabine. The study was positive for the primary endpoints; the results demonstrated a statistically significant improvement in PFS (HR 0.76; 95% CI: 0.63, 0.93; *p* = 0.0059) in favor of the neratinib plus capecitabine combination, with a mOS of 21 months vs. 18.7 months and an ORR of 32.8% vs. 26.7% in favor of the experimental arm. In terms of adverse effects, diarrhea was the most frequent AE in both arms. In the neratinib arm, it occurred in 83%, with grade 3 in 24.4%. In the lapatinib arm, it was present in 66%, with grade 3 reporting 12.5%. Antidiarrheal medication with loperamide has become an important support for prophylaxis with this type of agent [49].

Based on the results achieved in the NALA clinical trial, neratinib obtained FDA approval as a third line of treatment for HER2-positive metastatic breast cancer [50].

#### 2.3.3. Tucatinib

Tucatinib is a highly selective reversible inhibitor of the HER2 tyrosine kinase. In the HER2CLIMB trial, it was compared with placebo, both arms in combination with trastuzumab and capecitabine, in 612 patients previously treated with trastuzumab, pertuzumab, and T-DM1. Patients were randomized 2:1 in favor of tucatinib. The study included patients with treated and untreated brain metastases (BM) that did not need immediate local therapy. Previously treated brain metastases could be either stable since treatment or to have progressed since prior local CNS therapy, provided there was no clinical indication for immediate re-treatment with local therapy [51].

The PFS at 1 year, in the HER2CLIMB trial, was 33.1% in the tucatinib-combination group and only 12.3% in the placebo-combination group (HR 0.54, 95% CI0.42–0.71, *p* < 0.001). In patients with brain metastases, the PFS at 1 year was 24.9% for the tucatinib group vs. 0% for the placebo group (HR 0.48, 95% CI 0.34–0.69. *p* < 0.001). At a median follow-up of 29.6 months, the median duration of OS was 24.7 months for the tucatinib group and 19.2 months for the placebo group (HR: 0.73, 95% CI: 0.59–0.90, *p* = 0.004). The most common adverse events included diarrhea, palmar-plantar erythrodysesthesia syndrome, nausea, fatigue, and vomiting for the tucatinib-combination arm; just 5.7% of the patients had to discontinue tucatinib [52].

The HER2CLIMB trial included a total of 291 patients with brain metastases: 198 patients in the tucatinib arm (48%) and 93 in the control arm (46%). The risk of intracranial progression or death decreased by 68% for the experimental arm (HR: 0.32; 95% CI, 0.22–0.48; *p* < 0.0001). Median intracranial progression or death (CNS-PFS) was 9.9 months versus 4.2 months in favor of tucatinib. The mOS was 18.1 months vs. 12 months, also in favor of the tucatinib arm. The results obtained in this study with the combination of tucatinib, tastuzumab plus capecitabine led to FDA approval of tucatinib as a treatment option in the HER2-positive metastatic setting [53].

Recruitment is complete and results awaited for the HER2CLIMB-02 trial (NCT03975647), a phase III study that will evaluate efficacy and safety of tucatinib or placebo plus T-DM1 in patients with unresectable locally advanced or metastatic HER2-positive BC previously treated with taxane and trastuzumab in any setting. Another study recruiting is the HER2CLIMB-05 trial (NCT05132582), which will evaluate the addition of tucatinib to standard-of-care maintenance in the first-line setting.

For patients with HER2-positive early BC and invasive residual disease after neoadjuvant therapy, an ongoing, prospective, phase III trial (CompassHER2 RD [NCT04457596]) will evaluate the addition of tucatinib to T-DM1 in preventing relapse in this high-risk population.

#### 2.3.4. Pyrotinib

Pyrotinib is a second-generation, irreversible pan-HER TKI (HER1, HER2, and HER4). A phase I trial with 38 patients with HER2-positive metastatic breast cancer without previous exposure to HER2 tyrosine kinase inhibitors showed an ORR of 50% and median PFS of 35.4 weeks in 36 evaluable heavily pretreated metastatic patients and tolerable adverse effects with diarrhea grade 3 in two patients [54].

The efficacy and safety of pyrotinib or lapatinib in combination with capecitabine was compared in a phase II trial that included patients previously treated with taxanes and anthracyclines, with or without previous trastuzumab. A total of 128 patients were enrolled. The combination with pyrotinib significantly improved the ORR (78.5% vs. 57.1%) and PFS (18.1 vs. 7.0 months and HR 0.36, 95% CI 0.23–0.58) compared with those patients treated with lapatinib plus capecitabine. The most frequent grade 3–4 adverse events were hand-foot syndrome (24.6% vs. 20.6%); diarrhea (15.4% vs. 4.8%) and decreased neutrophil count (9.2% vs. 3.2%), respectively [55].

The PHOEBE study was a phase III trial, in which pyrotinib was compared to lapatinib, both in combination with capecitabine. It enrolled 267 patients with HER2-positive metastatic breast cancer who had previously received treatment with trastuzumab and taxanes. The results of this trial were positive in favor of the experimental arm, achieving an mPFS of 12.5 months vs. 6.8 months for the control arm (HR 0.39. CI 95% 0.27–0.56; *p* < 0.0001).

Concerning adverse effects, 10% of patients in the pyrotinib group and 8% in the lapatinib group had severe AEs; the most frequent grade 3 AEs were diarrhea and hand-foot syndrome [56]. Pyrotinib could be a treatment option in patients who have progressed to trastuzumab and chemotherapy; however, these studies have not enrolled patients who underwent treatment with multiple HER2-targeted drugs, and pyrotinib has not been compared with newer ADCs such as T-DXd.

### 2.4. Combinations to Overcome Resistance

#### CDK4/6 and PI3K-AKT Inhibitors

The HER2–ligand interaction with the PI3K-AKT pathway activates the cyclin D1/CDK4/6/pRb axis, and downstream activation of cyclin D1 can induce resistance to anti-HER2 therapies. On this preclinical basis, the combination of anti-HER2 therapies with CDK4/6 inhibitors (CDK4/6i) had been explored in different trials as an alternative strategy to overcome resistance [57,58].

In a phase I trial of patients with advanced solid tumors, abemaciclib showed activity in 11 patients with HER2-positive MBC enrolled in the study, with 36% of these patients achieving a partial response [59]. With these earliest observations, further evaluation of CDK4/6i treatment in HER2-positive BC has been considered. 

The MonarcHER, a phase II trial, was designed to test the combination of abemaciclib plus trastuzumab with or without fulvestrant compared to trastuzumab plus chemotherapy in patients with HER2-positive/HR positive MBC who had received at least two prior anti-HER2 treatments. 

Patients were included in three arms: Group A received the triplet abemaciclib, trastuzumab and fulvestrant; Group B was treated with abemaciclib and trastuzumab; and Group C received chemotherapy and trastuzumab. The primary objective was PFS, first testing group A versus group C. With 237 patients enrolled, the triplet combination treatment significantly improved PFS with a median for group A of 8.3 months versus 5.7 months for group C ( HR 0.67, 95% CI 0.45–1.00; *p* = 0.051). With a tolerable safety profile, the most common grade 3–4 adverse event was neutropenia (A: 27%, B: 22% and C: 26%) [60].

The final data from the MonarcHER study were presented at ESMO 2022 after a median follow-up of 52.9 months. Overall survival results were reported in favor of the triplet, although this was a secondary endpoint. The mOS was 31.1 months in arm A vs. 20.7 months in arm C (HR 0.75, 95% CI 0.47–1.21, *p* = 0.243). The final PFS and safety results were consistent with these previous findings. One limitation of the study was the lack of an endocrine and trastuzumab control arm that would confirm the relative contribution of abemaciclib to the combination. These results support this combination as an alternative treatment option for patients with HR-positive, HER2-positive MBC [61]. 

The phase II PATRICIA trial examined the antitumor activity of palbociclib in combination with trastuzumab with or without endocrine therapy in pretreated patients with HER2-positive BC. Three cohorts were established: estrogen receptor (ER)-negative (cohort A), ER-positive (cohort B1), and ER-positive with letrozole (cohort B2); and the primary endpoint was PFS rate at 6 months (PFS6). A total of 71 patients were recruited. The PFS6 rate in cohorts A, B1, and B2 was 33.3% (5/15), 42.8% (12/28), and 46.4% (13/28), respectively. The most common grade 3–4 adverse events were neutropenia (66.4%) and thrombocytopenia (11.3%). As a secondary objective, the evaluation of the defined intrinsic subtypes by PAM50 found that luminal disease was independently associated with longer PFS compared with non-luminal disease (10.6 vs. 4.2 months median PFS; adjusted hazard ratio 1⁄4 0.40; P 1⁄4 0.003) [62]. A new cohort of patients is being recruited to test the hypothesis that in patients with HER2-positive/ER-positive BC and PAM50 luminal disease, the combination treatment is superior to physician’s choice.

Regarding T-DM1, in a phase I trial, 18 patients were treated in combination with palbociclib and showed that the combination was safe, tolerable, and active in previously treated HER2-positive patients with an ORR of 33% and median PFS of 6 months (95% CI 2.5–11.6) [63]. Similarly, ribociclib was explored in a phase Ib/II trial in combination with trastuzumab or T-DM1. No dose-limiting toxicities were observed, but no objective responses were achieved, and median PFS was 1.33 months [64].

The PATINA trial [NCT02947685] will evaluate the hypothesis that adding palbociclib to the anti-HER2 therapy plus endocrine maintenance therapy after first-line chemotherapy + anti-HER2 therapy might improve the outcome of HR-positive/HER2-positive MBC patients (NCT02947685). Several trials evaluating the combination of HER2-targeted therapies with CDK4/6i and endocrine therapy are under way. Currently ongoing trials are listed in Table 2.

As the PI3K-AKT pathway is also implicated in resistance to HER2-targeted therapies, this insinuates a rationale for combining PI3K inhibitors (PI3Ki) and HER2-targeted therapy. A phase Ib trial tried to determine the clinical activity of the PI3Ki buparlisib in patients with HER2-positive MBC. With 100 mg/day as the declared recommended phase II dose (RP2D) of buparlisib in combination with trastuzumab, the mainly adverse events presented in patients were rash (39%), hyperglycemia (33%), and diarrhea (28%), and the disease control rate was 75%. However, in the phase II portion, despite a manageable toxicity profile, the combination of buparlisib plus trastuzumab did not reach the primary endpoint of ORR ≥ 25% [65,66]. 

Another trial reported was the phase Ib PantHER study that evaluated the safety of copanlisib (a pan-class I PI3K inhibitor with particular activity against PI3Kα) in combination with trastuzumab. The trial enrolled twelve pretreated patients, and there were no DLTs. Grade 3 adverse events were lung infections (n = 4), abdominal pain (n = 4), liver enzyme rise (n = 1) and hypertension (n = 4). Best response was stable disease in nine patients [67]. The efficacy of the combination will be assessed in a phase II study. 

Alpelisib and T-DM1 showed promising activity with a median PFS of 8.1 months and ORR of 30% in 17 trastuzumab and taxane-resistant HER2-positive MBC patients [68]. Recruiting is ongoing in a phase III study to evaluate the efficacy and safety of alpelisib compared to placebo in combination with anti-HER2 therapy as maintenance treatment of patients with HER2-positive MBC who present a PIK3CA mutation in tumor (Table 2). 

Pilaralisib, a selective, reversible, pan-class I PI3Ki, was investigated in a phase I/II study in combination with trastuzumab (Arm 1) or trastuzumab plus paclitaxel (Arm 2). There was some activity in the arm with paclitaxel, and correlation between PIK3CA mutations in cell-free circulating DNA and response were not observed [69].

Another targeted therapy under study was refametinib, an allosteric MEK1/2 inhibitor that showed, in combination with copanlisib and anti-HER2-targeted therapies, an antiproliferative effect when was used in HER2-positive breast cancer cell lines with acquired trastuzumab or lapatinib resistance [70]. 

Inavolisib is a new-generation PI3K alpha inhibitor that is currently being evaluated in combination with trastuzumab and pertuzumab in patients harboring PIK3CA mutations, as well as in the early neoadjuvant setting combined with endocrine treatment (NCT05306041). Currently ongoing trials with PI3Ki and HER2-targeted therapy are listed in Table 3.

### 2.5. Immune Approaches

#### Immune Checkpoint Inhibitors

Analysis of data from molecular studies of tumor microenvironment have shown that the greatest immunogenic potential is presented by the triple-negative breast cancer (TNBC) and HER2-positive BC subtypes [71]. Based on these data, immunotherapy appears to be an appealing strategy for these patients. However, prospective data available in this setting are scanty, and there are no immunotherapy single agent-trials specifically in a pure cohort of HER2-positive BC patients. 

A total of 168 patients, of whom 15.5% had HER2-positive MBC, were treated with Avelumab monotherapy in a phase Ib trial. The ORR was 3% overall, but none were reported in the HER2-positive cohort [72]. In another phase Ib/II trial, the combination of pembrolizumab with trastuzumab was explored in 58 HER2-positive MBC patients who had progressed to trastuzumab. The mean duration of disease control was 11.1 months, with 15% of patients in the PD-L1-positive group achieving an ORR, and 8% maintained a stable disease for more than 6 months. A subgroup analysis showed higher levels of tumor infiltrating lymphocytes (TILs) in the PD-L1-positive group, suggesting that the immune mechanism may be important in resistance to anti-HER2 therapy. There were no objective responders in the PD-L1-negative group [73].

The KATE2 trial evaluated the combination of atezolizumab plus T-DM1 in patients with locally advanced or metastatic HER2-positive BC after treatment with trastuzumab and taxanes. The primary endpoint was PFS. A median PFS of 8.2 months in the atezolizumab group versus 6.8 in the T-DM1 plus placebo group (HR 0.82, *p* = 0.33) and no differences in ORR between the two arms were reported. Although with no statistical significance, patients with PD-L1 positive and/or TILs ≥ 5%, had longer PFS and better ORR with atezolizumab. Thrombocytopenia (13% of patients who received atezolizumab vs. 4% who received placebo), elevated hepatic enzymes (13% vs. 6%), anemia (5% vs. 0), and neutropenia (5% vs. 4%) were the common grade 3 or worse adverse events [74]. Derived from this evidence, a phase III KATE3 study was initiated and is currently recruiting [75]. Moreover, the ASTEFANIA trial is exploring the addition of atezolizumab to T-DM1 in the adjuvant setting of patients with residual disease after neoadjuvant treatment (NCT04873362).

Several trials using immune checkpoint inhibitors (PD-1/PD-L1, CTLA-4) and/or combinations with anti-HER2 therapy are ongoing. Other target molecules such as CD137 (expressed on activated T cells and NK cells and up-regulated by trastuzumab) are under investigation. Currently ongoing trials with immunotherapy in HER2-positive BC are listed in Table 4.

## 3. Conclusions

In recent years, we have witnessed how a better understanding of tumor biology and resistance mechanisms has promoted the development of new targeted drugs more specific and potent, as HER2-positive breast cancer is one example of the most successful subtypes to benefit from these strategies. 

In the metastatic setting, the dual anti-HER2 blockade with trastuzumab and pertuzumab in the first line remains the preferable option and currently, T-DXd has become the unquestionable new standard treatment in the second line, which has drastically improved the natural history of these patients. From the third line of treatment, there are increasing options, and to decide the best strategy, we must consider the toxicity and efficacy of each treatment as well as the clinical characteristics of the patients (brain metastases, response to previous therapies, toxicities developed and biomarker analysis). However, despite these important therapeutic advances, most patients eventually relapse and become resistant. In response to this need, numerous new treatments and combinations with various mechanisms of action are being investigated, several of which we have mentioned and are promising options. As we mentioned above, there is growing evidence of mechanisms and possible cross-resistance, as well as a better knowledge and management of characteristic toxicities to these novel targeted agents and combinations, which will also aid in better defining the most convenient sequence for each patient.

There is solid data that support the use of anti-HER2 treatments in the (neo)adjuvant context, with an adequate risk–benefit balance (mainly cardiac toxicity) and a significant decrease in relapses and impact on survival. In addition, there are several trials ongoing with novel treatment strategies that include endocrine, targeted therapies or immune approaches in combination anti-HER2 therapies in this early (neo)adjuvant scenario. As these new therapies reach the early disease setting, it is essential to determine the impact on subsequent treatments, because relapses on treatment or after (neo)adjuvant treatment constitute an unfavorable scenario due to possible de novo or acquired resistances to these therapies, currently used in the metastatic setting. Unfortunately, there is a lack of consensus or high quality data regarding the issue of a treatment-free interval or treatment sequencing.

## Figures and Tables

**Table 1 cancers-15-00051-t001:** Other HER2-targeted ADCs in clinical development.

ADC	Antibody	Payload	DAR	Trials Ongoing—Results
ALT-P7	Trastuzumab variant	Monomethyl auristatin E	2	Dose-escalation phase 1 trial, ASCO 2020, #3551 PFS: 6.2 months
BDC-1001	Trastuzumab biosimilar	TLR7/8	NA	Phase I–II ongoing ASCO 2021, #2549
A-166	Trastuzumab	Auristatin based-Duo 5	NA	Phase II ongoing ASCO 2021, #1024
FS-1502 IKS014	Trastuzumab	Monomethyl auristatin F	NA	Phase I ongoing NCT03944499
Disitamab Vedotin RC48	Hertuzumab (anti-HER2 humanized Ab)	Monomethyl auristatin E	4	SABCS 2019, #PD4-06 ORR; 46.7%
ZW-49	ZW-25(Anti-HER2 bispecific Ab)	N-acyl sulfonamide auristatin	NA	Phase I ongoing NCT03821233 ESMO 2022, 460MO
MRG-002	Anti-HER2 Ig G1	Monomethyl auristatin E	NA	Phase I–II ongoing NCT04941339

ADC: Antibody-drug conjugate; DAR: Drug-to-antibody ratio; NA: Not available,

#: number of the abstract presented

**Table 2 cancers-15-00051-t002:** Currently ongoing trials with CDK4/6i.

Identifier	Phase	Drug	Endpoint	
NCT02947685	III	Palbociclib + anti-HER2 + ET	PFS	Active, not recruiting
NCT03530696	II	T-DM1 + Palbociclib	PFS	Active, not recruiting
NCT03304080	I/II	Anastrozole + Palbociclib + anti-HER2	DLT + RP2D	Recruiting
NCT03709082	I/II	TDM-1 + Palbociclib + Letrozole	DLT + RP2D	Active, not recruiting
NCT03054363	I/II	Tucatinib + Palbociclib + Letrozole	DLT + RP2D/PFS	Active, not recruiting
NCT03065387	I	Neratinib + Everolimus + Palbociclib or Trametinib	DLT + RP2D	Recruiting

ET: Endocrine Therapy; PFS: Progression-Free Survival; DLT: Dose Limiting Toxicities RP2D: Recommended Phase 2 Dose. ClinicalTrials.gov.

**Table 3 cancers-15-00051-t003:** Currently ongoing trials with PI3K/AKTi.

Identifier	Phase	Drug	Endpoint	
NCT04208178	III	Trastuzumab + Pertuzumab +/− Alpelisib	DLT + PFS	Recruiting
NCT05063786	III	Trastuzumab + Alpelisib +/− Fulvestrant vs. Trastuzumab + Chemotherapy	PFS	Recruiting
NCT04108858	I/II	Copanlisib + Pertuzumab + Trastuzumab	DLT + RP2D/PFS	Recruiting
NCT04253561	I	Ipatasertib + Pertuzumab + Trastuzumab *	DLT + RP2D	Recruiting
NCT05230810	I/II	Alpelisib + Tucatinib + Fulvestrant	DLT + RP2D/PFS	Not yet recruiting
NCT04253561 *	I	Ipatasertib + Trastuzumab + Pertuzumab *	DLT + RP2D	Recruiting
NCT02390427	I	Taselisib + Pertuzumab + Trastuzumab +/− T-DM1	DLT + RP2D	Active, not recruiting
NCT03767335	I	MEN1611 + Trastuzumab +/− Fulvestrant	DLT + RP2D	Active, not recruiting
NCT03006172 ç	I	Inavolisib + Pertuzumab + Trastuzumab	DLT + RP2D	Recruiting

PFS: Progression-Free Survival; DLT: Dose Limiting Toxicities; RP2D: Recommended Phase 2 Dose. ClinicalTrials.gov. * Presented as a poster at SABCS 2021, with a median follow-up of 4 months, 6 patients were treated with Ipatasertib (400 mg orally once daily D1-21 q28d) + Trastuzumab + Pertuzumab and presented as most common treatment-related AEs: diarrhea (n = 5, 83.3%) and nausea (n = 2, 33.3%), mostly grade 1. No DLTs or grade 3/4 AEs were observed. ç A two-stage trial with other combinations of Inavolisib with endocrine and targeted therapies in patients with locally advanced or metastatic PIK3CA-mutant BC.

**Table 4 cancers-15-00051-t004:** Currently ongoing trials with immunotherapy.

Identifier	Phase	Drug	Endpoint	
DIAmOND	II	Durvalumab + Tremelimumab + Trastuzumab	PFS	Active, not recruiting
NCT03414658	II	Trastuzumab + Vinorelbine +/− Avelumab and Utomilumab	PFS	Recruiting
NCT03199885	III	Paclitaxel + Trastuzumab + Pertuzumab +/− Atezolizumab	PFS	Active, not recruiting
NCT03125928	II	Paclitaxel + Trastuzumab + Pertuzumab + Atezolizumab	TAE	Recruiting
NCT02605915	Ib	Atezolizumab Combination Treatments	DLT + RP2D	Active, not recruiting
NCT03417544	II	Atezolizumab + Trastuzumab + Pertuzumab	ORR in CNS	Active, not recruiting
NCT03364348	I	Utomilumab + Trastuzumab OR T-DM1	DLT + RP2D	Active, not recruiting
NCT03032107	I	Pembrolizumab + T-DM1	DLT + RP2D	Active, not recruiting
NCT03650348	I	PRS-343 + Atezolizumab	DLT + RP2D	Active, not recruiting
NCT03330561	I	PRS-343	DLT + RP2D	Active, not recruiting
NCT02297698	II	Nelipepimut-S (vaccine) + Trastuzumab	DFS	Active, not recruiting

PFS: Progression-Free Survival; TAE: Treatment-related Adverse Event; EFS: Event-Free Survival; DFS: Disease-Free Survival; ORR: Overall Response Rate; DLT: Dose-Limiting Toxicities; RP2D: Recommended Phase 2 Dose; CNS: Central Nervous System. ClinicalTrials.gov.

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
