# Peer review of "Systemic Therapy for HER2-Positive Metastatic Breast Cancer: Current and Future Trends"

_cancers, 2022, doi:10.3390/cancers15010051_

Round 1

Reviewer 1 Report

The manuscript by Vega Cano et al presents a review on the therapies for HER2-positive metastatic breast cancer. The paper systematically reviewed treatment options for advanced Her+ breast cancer, which span from standard protocols to ongoing trials. The merit of the manuscript is its systematic approaches in the introduction of various Her2+ targeting agents in context with clinical outcomes and adverse events from specific clinical trials. It is a concise summary of the relevant therapies, which will be a useful reference for readers to following up proceedings in Her2 targeted therapy. However, The manuscript could be strengthened by including more discussion on integrated results from different trials to reflect some mechanistic insight.

Author Response

Dear Editor and Reviewers.

Following our previous correspondence, we wish to submit a point-by-point rebuttal to the reviewer’s comments on our original research article entitled “Systemic therapy for HER2-positive Metastatic Breast Cancer: Current trends and Future Perspective”. We provide a comprehensive explanation regarding how the new version of the manuscript addresses these comments.

We hope that the rebuttal is satisfactory, but please do not hesitate to contact me should you need any additional clarification.

Thank you in advance for considering a resubmission of this manuscript.

Warm regards,

Reviewer 2 Report

This is a good review of an important subject. One important point that has been missed is the impact that (neo) adjuvant therapies have on the disease that later recurs; although pertuzumab and T-DM1 have had a huge impact in reducing relapses, those that happen are more likely to be already resistant to these lines of therapy. I would suggest that this is added to the abstract and discussion.

A figure summarising recommended treatment paradigms from the results you have discussed could be usefully added.

Some additional points;

1.       Margetuximab; the lack of a companion diagnostic prevents the selection of patients with the CD16A 158F allele, which makes this somewhat obsolete given the lack of benefit in the overall population compared to trastuzumab (which is cheaper and can be delivered sc)

2.       TDM1: Given the widespread use of post-neoadjuvant TDM1 after non-pCRs, the Katherine trial deserves discussion here including the reduction in risk of extra-cranial relapse, thereby increasing the proportion of relapsers with intracranial disease.

3.       ARX788; there is a typo in the heading. You have stated that the pharmaceutical company is no longer developing this agent, yet trials are ongoing in CHina. Please can you reference this information and comment on why, if known.

4.       Lapatinib; please comment that prior to the EMILIA trial, the cape/lapatinib combination was the standard 2nd line regimen internationally

5.       Neratinib; please add discussion of TBCRC 022 and the impressive ORR in brain metastases

6.       Pyrotinib; final sentence should say TDXD rather than TDM1 I think?

7.       Combinations; the major limitation of Monarch-HER was the lack of a proper control arm (fulvestrant/trastuzumab) as without that, the relative contribution of abemaciclib is completely unknown

8.       Combinations line 567; “Table 1” should read “Table 2”

9.       Combinations; Inavolisib is also being evaluated in the neo-adjuvant and maintenance setting (NCT 03006172, NCT05306041), so should also be discussed and added to table 3 please

10.   Immune approaches: You could also mention the adjuvant ASTEFANIA trial here

This is a good review of an important subject. One important point that has been missed is the impact that (neo) adjuvant therapies have on the disease that later recurs; although pertuzumab and T-DM1 have had a huge impact in reducing relapses, those that happen are more likely to be already resistant to these lines of therapy. I would suggest that this is added to the abstract and discussion.

A figure summarising recommended treatment paradigms from the results you have discussed could be usefully added.

Some additional points;

1.       Margetuximab; the lack of a companion diagnostic prevents the selection of patients with the CD16A 158F allele, which makes this somewhat obsolete given the lack of benefit in the overall population compared to trastuzumab (which is cheaper and can be delivered sc)

2.       TDM1: Given the widespread use of post-neoadjuvant TDM1 after non-pCRs, the Katherine trial deserves discussion here including the reduction in risk of extra-cranial relapse, thereby increasing the proportion of relapsers with intracranial disease.

3.       ARX788; there is a typo in the heading. You have stated that the pharmaceutical company is no longer developing this agent, yet trials are ongoing in CHina. Please can you reference this information and comment on why, if known.

4.       Lapatinib; please comment that prior to the EMILIA trial, the cape/lapatinib combination was the standard 2nd line regimen internationally

5.       Neratinib; please add discussion of TBCRC 022 and the impressive ORR in brain metastases

6.       Pyrotinib; final sentence should say TDXD rather than TDM1 I think?

7.       Combinations; the major limitation of Monarch-HER was the lack of a proper control arm (fulvestrant/trastuzumab) as without that, the relative contribution of abemaciclib is completely unknown

8.       Combinations line 567; “Table 1” should read “Table 2”

9.       Combinations; Inavolisib is also being evaluated in the neo-adjuvant and maintenance setting (NCT 03006172, NCT05306041), so should also be discussed and added to table 3 please

10.   Immune approaches: You could also mention the adjuvant ASTEFANIA trial here

Author Response

Dear Editor and Reviewers.

Following our previous correspondence, we wish to submit a point-by-point rebuttal to the reviewer’s comments on our original research article entitled “Systemic therapy for HER2-positive Metastatic Breast Cancer: Current trends and Future Perspective”. We provide a comprehensive explanation regarding how the new version of the manuscript addresses these comments.

We hope that the rebuttal is satisfactory, but please do not hesitate to contact me should you need any additional clarification.

Thank you in advance for considering a resubmission of this manuscript.

Best regards,
